# Design of Hybrid Quinoline–Chalcone Compounds Against *Leishmania amazonensis* Based on Computational Techniques: 2D- and 3D-QSAR with Experimental Validation

**DOI:** 10.3390/ph18101567

**Published:** 2025-10-17

**Authors:** Marcos Lorca, Gisela C. Muscia, Jaime Mella, Luciana Thomaz, Jenicer K. Yokoyama-Yasunaka, Daniel Moraga, Yeray A. Rodriguez-Nuñez, Silvia E. Asís, Mauro Cortez, Marco Mellado

**Affiliations:** 1Facultad de Ciencias de la Vida, Carrera de Química y Farmacia, Universidad Viña del Mar, Viña del Mar 2572002, Chile; marcos.lorca@uvm.cl; 2Departamento de Ciencias Químicas, Facultad de Farmacia y Bioquímica, Universidad de Buenos Aires, Junín 956, Ciudad Autónoma de Buenos Aires, Buenos Aires 1113, Argentina; elizabet@ffyb.uba.ar; 3Instituto de Química, Facultad de Ciencias, Universidad de Valparaíso, Av. Gran Bretaña 1111, Valparaíso 2360102, Chile; jaime.mella@uv.cl; 4Centro de Investigación, Desarrollo e Innovación de Productos Bioactivos (CInBIO), Universidad de Valparaiso, Av. Gran Bretaña 1111, Valparaíso 2360102, Chile; 5Institute of Biomedical Sciences, Department of Parasitology, University of São Paulo, Av. Prof. Lineu Prestes 1374, São Paulo 05508-220, Brazil; lucithomaz2016@gmail.com (L.T.); jenicerk@usp.br (J.K.Y.-Y.); 6Laboratorio de Fisiología, Departamento de Ciencias Biomédicas, Facultad de Medicina, Universidad de Tarapacá, Arica 1000000, Chile; dmoraga@academicos.uta.cl; 7Laboratorio de Síntesis y Reactividad de Compuestos Orgánicos, Departamento de Ciencias Quimicas, Facultad de Ciencias Exactas, Universidad Andrés Bello, Santiago 8370146, Chile; yeray.rodriguez@unab.cl; 8Escuela de Tecnología Médica, Facultad de Ciencias, Pontificia Universidad Católica de Valparaíso, Valparaíso 2373223, Chile; 9Centro de Investigación en Ingeniería de Materiales, Universidad Central de Chile, Santiago 8330507, Chile

**Keywords:** leishmaniasis, drug design, 2D-QSAR model, 3D-QSAR model, comparative molecular similarity index Analysis-CoMSIA, quinoline–chalcone synthesis, experimental validation

## Abstract

**Background**: *Leishmania amazonensis*, one of the causative agents of cutaneous leishmaniasis, is responsible for a neglected tropical disease affecting nearly one million individuals worldwide. Although clinical treatments are available, their effectiveness is often compromised by high toxicity and limited selectivity. **Methods**: From a database, 64 chalcone derivatives were studied using Comparative Molecular Similarity Indices Analysis (CoMSIA) and Hansch analysis (2D-QSAR) to construct predictive computational models. Internal and external validation criteria were applied to identify structural determinants associated with antileishmanial activity. Based on these insights, twelve novel quinoline–chalcone hybrids were designed, synthesized, and biologically evaluated against *L. amazonensis*. **Results**: The most robust CoMSIA model combined steric and hydrogen-bond acceptor fields (CoMSIA-SA), while Hansch analysis highlighted electronic descriptors—specifically LUMO energy and the global electrophilicity index—as critical for parasite growth inhibition. Guided by these computational findings, a new series of 12 hybrid quinoline–chalcone derivatives (**E001**–**E012**) was synthesized through a two-step procedure, achieving overall yields of 43–71%. Biological assays demonstrated that four compounds displayed inhibitory activity comparable to amphotericin B. Furthermore, physicochemical profiling and in silico pharmacokinetic predictions for the most active compounds (**E003**, **E005**, **E006**, and **E011**) indicated favorable biocompatibility and drug-like properties. **Conclusions**: These results underscore the value of an integrative computational–experimental approach in the rational design of next-generation *L. amazonensis* inhibitors, reinforcing chalcone-based scaffolds as promising pharmacophoric frameworks for antileishmanial drug discovery.

## 1. Introduction

Leishmaniasis is a zoonotic neglected disease caused by protozoan parasites of the genus *Leishmania*, transmitted through the bite of infected sandflies. This disease presents several clinical forms, including cutaneous, mucocutaneous, and visceral leishmaniasis, with the latter being the most severe and associated with a high mortality rate if not treated properly [1,2]. It is estimated that there are between 700,000 and 1 million new cases annually. The disease affects some of the poorest populations on the planet and is associated with factors such as biological conditions (malnutrition and weakened immune system) and social conditions (population displacement, poor housing conditions, and lack of economic resources [3,4].

The current treatment of leishmaniasis varies depending on the clinical form of the disease, the *Leishmania* species involved, the geographic region, and the patient’s condition. Leishmaniasis is a complex parasitic disease that presents therapeutic challenges, mainly due to resistance to conventional medications, treatment toxicity, and limitations in available options in some endemic areas [5,6,7]. For example, the visceral leishmaniasis (VL), treatment includes the use of pentavalent antimonials (such as **meglumine antimoniate** and **pentostam**) or liposomal **amphotericin B** [8], with the lowest toxicity than antimonial derivatives, especially in cases of resistance or toxicity to antimonials, and in immunocompromised patients [9]; **miltefosine** is used to treat the VL and cutaneous leishmaniasis (CL) [10], as well as topical imidazoles like **miconazole** and **ketoconazole** with activity against the parasite from in vitro studies [11]. All compounds used in the treatment are shown in Figure 1.

Despite the conventional pharmaceutical treatment being available, resistance to drugs is a problem [12,13]. For this reason, the scientific community is researching new compounds with antileishmanial activity, founding the quinoline core (**I**) as a potential alternative [14], among them, 2-styrylquinoline-4-carboxylic acid (**I-1**) [15], 7-trifluoromethyl-2-phenyl-4-quinoline carboxylic acid (**I-2**) [16], and 6-chloro-2-methyl-4-phenylquinolines (**I-3**) [17]. In addition, it has been demonstrated that certain chalcones can inhibit the growth of *Leishmania* spp. [18,19]. For example, the natural compound licochalcone (**II-1**) shows inhibition activity on *Leishmania infantum chagasi* (species associated with VL), and the synthetic derivative **II-2** has inhibition activity on several *Leishmania* strains through blocking the cytosolic enzyme tryparedoxin peroxidase (cTXNPx) [20]. The chemical formulations of the quinoline and chalcone compounds mentioned before are shown in Figure 2.

An emerging strategy in the development of highly active compounds is the combination of two or more molecular fragments in a unique structure, obtaining a compound known as a hybrid compound, which can lead to improved pharmacological performance [21,22]. In this context, the construction of structure–activity relationship (QSAR) models, in both two and three dimensions, has become a key tool for the rational design of novel bioactive candidates [23,24]. This computational approach not only significantly reduces the costs associated with drug development but also shortens the time required to identify lead compounds. Consequently, the integration of computational and synthetic approaches represents a promising pathway for the discovery of new chemical entities with pharmacological potential [25].

Due to the importance of finding and validating new compounds against diseases such as leishmaniasis, in this work we have developed two computational models based on three-dimensional quantitative structure-activity (3D-QSAR) using Comparative Molecular Similarity Index Analysis (CoMSIA). In addition, two-dimensional quantitative structure-activity (2D-QSAR or Hansch’s analysis) was developed based on quantum descriptors. Thus, a new series of quinoline–chalcone hybrids was designed and synthesized, and the products were further evaluated for their inhibitory activity to validate the model.

## 2. Results

### 2.1. Theoretical Models

After compiling the dataset of the inhibitory activity of chalcone derivatives against *Leishmania amazonensis*, a 3D-QSAR model was constructed using CoMSIA (Comparative Molecular Similarity Indices Analysis) to aid in the design of new structures with activity against this strain of *Leishmania*. This strategy allows the rational direction of chemical modifications and synthetic efforts, thereby reducing the cost of the synthetic stage in the search for a lead compound [26]. The theoretical model was developed using a dataset of thirty-seven compounds listed in Appendix A, all of which contain the chalcone pharmacophore. The reported IC_50_ values were converted to pIC_50_ values (−log_10_(IC_50_)), revealing an activity range of 3 logarithmic units (pIC_50_ max = 6.585, pIC_50_ min = 3.542).

To obtain the best-performing CoMSIA model, a systematic search was carried out to explore various combinations of molecular fields—steric, electrostatic, hydrophobic, hydrogen-bond donor, and acceptor (S, E, H, D, and A, respectively)—to identify those providing optimal statistical parameters (summarized in Appendix A). This table shows that sixteen models exceed the desired threshold for predictive power (Condition 1: q^2^ > 0.5).

However, an important criterion for selecting the most robust models is the use of a low number of components, to avoid overfitting. In this regard, as a rule of thumb, the optimal number of components is considered to be 1/10 of the dataset size used for model construction [27,28]. Among the models analyzed, the one combining steric and hydrogen-bond acceptor fields (CoMSIA-SA) displayed the highest q^2^ value with a low number of components (q^2^ = 0.664, N = 2). Additionally, the CoMSIA-SA model was externally validated using several statistical parameters such as r^2^_test_, r^2^_0_, k, r^2^_m_, among others, which are summarized in Appendix A.

Additionally, a *Y*-randomization test was performed to ensure that the statistical significance of the model was not due to chance [29]. All randomized models yielded q^2^ values below the threshold established in Condition 1 of Appendix A, including negative values (see Appendix A), indicating the robustness and reliability of the original model. For external validation, compounds **012**, **025**, **026**, and **031** were excluded from the training set and assigned to the test set (Figure 3). Chalcone derivatives **012** and **025** exhibited residual values higher than those of the remaining compounds (Figure 3B). However, despite derivatives **026** and **031** showing residuals within the range observed for the training set, their inclusion in the test set caused a significant drop in the r_0_’^2^ and k’ statistics. Finally, the experimental pIC_50_ values (−log_10_(IC_50_)) were plotted against the predicted values from the CoMSIA-SA model. Both training and test set compounds (excluding outliers) projected closely onto the dashed light gray line, corresponding to the identity function (*y* = *x*), indicating good predictive performance.

### 2.2. Contour Map Analysis

Based on the CoMSIA model obtained, contour maps were generated to visualize the steric and hydrogen-bond acceptor fields. These fields are represented by polyhedral regions of different colors, indicating areas that are either favorable or unfavorable for the biological activity under study—in this case, the inhibition of *L. amazonensis* growth. The contour maps of the CoMSIA-SA model were projected according to the statistical parameters shown in Appendix A. To illustrate the favorable and unfavorable regions, the most active compound (**008**, pIC_50_ = 6.585, see Appendix A) and the least active compound in the training set (**037**, pIC_50_ = 3.542, see Appendix A) were used, as shown in Figure 4.

#### 2.2.1. Steric Map

The steric contour map of the CoMSIA-SA model (Figure 4A,B) reveals two favorable regions, indicated by green polyhedra, surrounding both aromatic rings of the chalcone scaffold in compound **008** (Figure 4A). For the 3,4,5-trimethoxyphenyl ring, the green polyhedron is primarily projected over the methoxy substituents, which is consistent with the high inhibitory activity of this compound. However, adjacent to the favorable green region, a yellow polyhedron is observed, indicating that bulky substituents in this area would negatively impact activity. In this context, the aromatic ring directly bonded to the carbonyl group via a single bond has rotational freedom along its axis. This feature causes the minimum energy conformation of compound **037** to orient the ring away from the mentioned polyhedral regions (Figure 4B).

On the other hand, for the aromatic ring attached to the β-carbon of the chalcone, yellow polyhedra are projected on both the front and back faces of the benzene ring. In the case of the most active chalcone in the dataset (compound **008**), a green polyhedron is projected over the aromatic ring bearing the nitro group, suggesting that a bulky substituent in this region favors the inhibition of *L. amazonensis* growth. However, since this ring is also connected to the β-carbon of the chalcone through a single bond, the lowest-energy conformation of any derivative may result in this ring being oriented outside the polyhedral regions, as seen in the compound **008** (Figure 4B).

#### 2.2.2. Hydrogen-Bonding Acceptor Map

The hydrogen-bond acceptor contour map derived from the CoMSIA model reveals two key regions influencing the inhibitory activity against *L. amazonensis* (Figure 4C,D), corresponding to the carbonyl carbon and the benzene ring attached to the β-carbon of the chalcone scaffold. The first region is enclosed by a magenta polyhedron, indicating that the presence of a hydrogen-bond acceptor group is crucial for antileishmanial activity, in agreement with the chalcone pharmacophore core.

Additionally, red polyhedra are projected near the para positions of both aromatic rings, which is consistent with the reduced activity observed for the compound **008** (Figure 4D). A red polyhedron is also projected near the meta position of the benzene ring bonded to the β-carbon, aligning with the lower inhibitory activity of derivative **035** (pIC_50_ = 5.4711 M), which contains an isoprenyl group directed toward this unfavorable region. Conversely, a small magenta polyhedron is projected near the same meta position of the benzene ring, indicating that a hydrogen-bond acceptor substituent in this area would enhance the inhibitory activity against *L. amazonensis*. This is in agreement with the oxygen atom of the nitro group present in the most active compound of the dataset, **compound 008**, pIC_50_ = 6.585 (Figure 4C).

### 2.3. Development of the 2D-QSAR Model

To complement the structural information obtained from the 3D-QSAR analysis, a 2D-QSAR study was performed using steric, topological, and quantum descriptors. This approach eliminates the dependence on the conformational arrangement of the compounds, which is a fundamental requirement in 3D-QSAR studies [26]. The 2D-QSAR analysis revealed that the inhibitory activity against *L. amazonensis* is primarily influenced by the energy of the LUMO (Lowest Unoccupied Molecular Orbital) and the global electrophilicity index (ω), as shown in Equation (1).pIC_50_ = 33.61**LUMO** + 7.52**LUMO^2^** + 17.43**ω** − 1.79**ω^2^** + 1.02(1)N = 22; r = 0.886; r^2^ = 0.785; SD = 0.051; F = 15.51; q^2^ = 0.726; r^2^_Test_ = 0.644

The results of the 2D-QSAR model for the inhibitory activity against *L. amazonensis* are shown in Figure 5 and Appendix A. Comparing the statistical outcomes of the developed 2D-QSAR model with the standard validation criteria for QSAR models (r^2^, q^2^, r^2^_test_), it is evident that the training set fits well to the ideal *y* = *x* line. This observation is consistent with the obtained r^2^ and q^2^ values, both exceeding the thresholds required for model validation. Additionally, the statistical performance of the test set (r^2^_test_ = 0.644) is considered acceptable.

From a physical perspective, the descriptors included in Equation (1) can be interpreted as follows: the LUMO energy is associated with Lewis acidity, while the global electrophilicity index reflects the stabilization energy of a molecular system upon gaining an electronic charge [30]. Since, in the aforementioned equation, the most significant descriptor is the LUMO energy and its squared value, these were plotted in Figure 5, considering the most and least active compounds in the database.

Figure 6 illustrates that in the most active compound, the LUMO is mainly localized on the α,β-unsaturated carbonyl–benzene fragment of the chalcone scaffold, whereas in the least active compound, this molecular orbital is delocalized across the entire molecule (Figure 6A,B). Furthermore, when plotting the squared orbital (LUMO^2^), it is observed that in the most active compound, the electron density is mainly concentrated on the nitro group and the carbonyl oxygen atom, while in the least active compound, the LUMO^2^ is localized over the α,β-unsaturated carbonyl fragment (Figure 6C,D). When comparing this information with the results from the 2D-QSAR and CoMSIA analyses (Figure 4 and Figure 6), a correlation can be seen between the LUMO^2^ distribution and the hydrogen-bond acceptor contour map. This is consistent with the physical interpretation of the LUMO, which acts as a Lewis acid capable of accepting electron pairs to form new bonds—analogous to hydrogen-bond formation. Interestingly, the projection of the LUMO^2^ in the most active compound overlaps with the magenta polyhedra in the CoMSIA hydrogen-bond acceptor model (Figure 4C and Figure 6C).

### 2.4. Summary of the Principal Results from the Computational Models and Design

Based on the development of the 3D-QSAR model using CoMSIA with steric and hydrogen-bond acceptor fields, the most relevant structural information for modulating the inhibitory activity against *L. amazonensis* was extracted. This information is summarized in Figure 7.

Based on the summary of the key chemical properties that modulate the inhibitory activity against *L. amazonensis*, and considering that both chalcone and quinoline pharmacophoric cores exhibit activity against this parasite, a new series of hybrid compounds is proposed by fusing these two scaffolds. In this design, the quinoline moiety would replace ring A of the chalcone, with the fusion occurring at positions R_3_ and R_4_ (Figure 8). This substitution is expected to enhance the inhibitory activity against *L. amazonensis*. Furthermore, as an extension of the model, the introduction of a bulky substituent at position R_6_ of the quinoline ring—corresponding to position R_3_ in the QSAR summary—is proposed to improve biological activity. Additionally, the model suggests that the incorporation of a hydrogen-bond acceptor at R_2_ position could further increase activity; therefore, a fluorine atom attached to a phenyl ring is proposed at position R_4_ of the quinoline core. This substitution is expected to orient the halogen atom toward the magenta polyhedron observed in Figure 4 through the slight rotational flexibility of the phenyl ring at this position (Figure 8). Lastly, the introduction of a small substituent at position R_2_ of the quinoline core (equivalent to position 6 of ring A in the chalcone scaffold) is also proposed for further optimization.

In addition, the α,β-unsaturated carbonyl group characteristic of chalcones is retained, along with ring B of benzene origin, which can be substituted with halogenated or methoxylated groups that may act as hydrogen-bond acceptors, primarily at position R_3_′. For position R_4_′ of ring B, the introduction of bulky substituents or groups with hydrogen-bond accepting capacity is proposed as a strategy to further modulate inhibitory activity.

### 2.5. Synthesis of New Quinoline–Chalcone Hybrids Driven by QSAR Models

To prepare the quinoline–chalcone hybrids, the starting material acetyl-quinoline (**III**) was synthesized from the 2′-fluorobenzophenone derivative (**III-1**), diketone (**III-2**) and acid catalysis employing the Friedländer reaction (Figure 1) [17]. Thus, the acetyl-quinoline (**III**) was subjected to the Claisen–Schmidt reaction under alkaline medium using different aromatic aldehydes (**IV**) with substituents having hydrogen-bond acceptor, and/or bulky characteristics to obtain a series of quinoline–chalcone hybrids (**E001**–**E012**).

After the quinoline–chalcone hybrids (**E001**–**E012**) synthesis, these were evaluated as growth inhibitory agents of *L. amazonensis* using the MTT assay according to the previous report using as positive control the antibiotic amphotericin B [15]. The results obtained were reported as half inhibition concentration (IC_50_), and converted to pIC_50_ (−log(IC_50_)). The results were compared with the theoretical values provided by the CoMSIA-SA and 2D-QSAR models (Table 1).

In terms of the predictive quality of the developed models, it can be observed that both models exhibit predictive power within the acceptable range, given that the residual value is less than 1.0 logarithmic unit. Based on these results, the CoMSIA-SA and 2D-QSAR models may be considered experimentally validated. Furthermore, the predicted results show that the CoMSIA-SA model displays greater precision than the 2D-QSAR model, despite the uneven contribution of steric and hydrogen-bond acceptor fields (77.5% and 22.5%, respectively). Regarding the contour map results, these are shown in Figure 9, including the most active compound **E006** and the least active one the compound **E002** among the quinoline–chalcone hybrids. This figure reveals that compound **E002** adopts a markedly different conformation compared to the most active compound (**E006**). In this regard, it has been reported that chalcones may adopt different conformations around the α,β-unsaturated carbonyl system which leads to a spatial reorientation of the substituents surrounding this moiety [31].

For the most active compound, the steric contour map shows that ring B is almost entirely projected onto the green polyhedron (Figure 9A), where the presence of a bulky substituent, including the chlorine atom bonded at position 4, is favored. Regarding ring A, it undergoes a slight torsion, losing coplanarity with the α,β-unsaturated carbonyl group and projecting part of the quinoline moiety into the green polyhedron, which enhances the inhibitory activity against *L. amazonensis*. Additionally, the methyl group at position 2 does not project into the yellow polyhedron due to the torsion of the chalcone’s ring A, a behavior similar to that of the 2-fluorophenyl fragment bonded to the quinoline ring at position 4. In contrast, for the least active compound among the quinoline–chalcone hybrids (**E002**), the minimum energy structure shows that the torsion of ring A positions it perpendicularly to the α,β-unsaturated carbonyl fragment, resulting in the quinoline moiety projecting into both yellow and green polyhedra (Figure 9B). As for ring B, it retains coplanarity with the α,β-unsaturated fragment; however, the conformational change in the minimum energy structure leads to part of the benzo[*d*][1,3]dioxole ring projecting into the yellow polyhedron, thereby reducing the inhibitory activity against *L. amazonensis*.

Regarding the hydrogen-bond acceptor field (Figure 9C,D), both the most and least active compounds (**E006** and **E002**, respectively) project the oxygen atom of the α,β-unsaturated carbonyl group into the magenta polyhedron, which enhances the inhibitory activity against *L. amazonensis*. Nevertheless, in the less active compound (**E002**), the B ring orients the benzo[*d*][1,3]dioxole fragment toward the red polyhedron (Figure 9D), suggesting that a substituent with hydrogen-bond acceptor characteristics in this region diminishes the inhibitory activity against the parasite. This effect is not observed in the most active compound (**E006**), which positions halogen atoms near this polyhedron (Figure 9C).

### 2.6. Assessment of Physicochemical Profiles and ADME Properties

The physicochemical and pharmacokinetic properties were evaluated using the online SwissADME platform (http://www.swissadme.ch/index.php (accessed on 8 March 2025)) for the most active compounds (Table 2). The quinoline–chalcone hybrid **E006**, which exhibited the highest activity (IC_50_ = 2.23 μM), possesses three hydrogen-bond acceptors, in contrast to compounds **E003**, **E005**, and **E011**, which contain a greater number of heteroatoms (e.g., nitrogen, oxygen, fluorine) capable of acting as hydrogen-bond acceptors. As discussed in the hydrogen-bond acceptor contour map analysis, the presence of such substituents may reduce the inhibitory activity against the parasite, particularly when located near position 4 of the B ring in the chalcone moiety (Figure 9C).

Moreover, the molar refractivity and lipophilicity of the most active quinoline–chalcone hybrid against *L. amazonensis* exhibited the highest values, which correlates with the importance of the steric field identified in the 3D-QSAR analysis (77.5% contribution from the steric field). Regarding water solubility (ESOL log S and ESOL class), the compounds showed low solubility, suggesting that their administration may require formulation in a lipophilic solution. Additionally, the four most active compounds demonstrated low intestinal absorption and a low probability of crossing the blood–brain barrier, thereby reducing the likelihood of central nervous system-related side effects. Another characteristic of these quinoline–chalcone hybrids is the absence of predicted inhibition of cytochrome enzymes CYP2C19, CYP2C9, CYP2D6, and CYP3A4, with the exception of compounds **E003** and **E011**, which may potentially inhibit CYP2C19, an enzyme involved in hepatic metabolism. The compounds exhibited a single violation of Lipinski’s rule of five, specifically due to logP values exceeding 5. This physicochemical characteristic is associated with a reduced oral bioavailability, with a predicted score of 0.55 for each compound. The predicted oral bioavailability (0.55) suggests a moderate absorption profile, which, although not optimal, may be sufficient for therapeutic efficacy. Further formulation strategies (e.g., the formulation of nanoparticles) could enhance systemic exposure, especially considering the lipophilic nature of the compounds.

The prediction that the compounds act as substrates of P-glycoprotein (P-gp) suggests a potential limitation in their oral bioavailability and intracellular accumulation, which could reduce their effectiveness as inhibitors of *L. amazonensis*. This efflux transporter, known for its role in drug expulsion, may remove the compounds before they exert their antiparasitic activity. However, this characteristic could also be advantageous in terms of limiting exposure to the central nervous system, thereby reducing the risk of neurological side effects. Consequently, further in vitro and in vivo studies should assess the impact of P-gp interaction on the therapeutic efficacy and tissue distribution of these quinoline–chalcone hybrids.

## 3. Materials and Methods

### 3.1. Computational Models

#### 3.1.1. Development of 3D-QSAR Model

The three-dimensional Quantitative Structure-Activity Relationship model was constructed using the SYBYL-X 1.2 software package (Tripos Inc., St. Louis, MO, USA), employing the Comparative Molecular Similarity Indices Analysis (CoMSIA) methodology, as described in our previous study [32]. Briefly, a dataset of 37 bioactive compounds was compiled from literature reports by Zottis, Alonso, Duval, Rodrigues-Filho, Echeverria, and Magalhães [33,34,35,36,37,38]. Selection criteria included the *Leishmania amazonensis* strain, growth stage of the parasite, experimental assay type, and reference standards used (Appendix A). Biological activity was expressed as the negative logarithmic scale of IC_50_ (pIC_50_ = −log_10_(IC_50_), mol·L^−1^).

The database was randomly split as follows: 25 compounds, 68% of training set and 12 compounds, 32% of a test set. The distribution of pIC_50_ values for the complete dataset, as well as for the training and test subsets, is illustrated in Appendix A.

Molecular structures were initially drawn using ChemDraw version 15.1.0.144 (PerkinElmer Inc., Shelton, CT, USA), and energy minimization was performed using the Tripos force field, including the Gasteiger–Hückel partial charges on each atom. All compounds were aligned atom-by-atom using the common α,β-unsaturated carbonyl fragment as a template (Appendix A).

#### 3.1.2. Comparative Molecular Similarity Index (CoMSIA) Field Generation

CoMSIA descriptors were obtained by embedding the aligned molecules of the training set into a three-dimensional cubic lattice with 2.0 Å spacing along the x, y, and z directions. The analysis employed the standard CoMSIA parameters, including a probe atom with a +1.0 charge, a radius of 1.0 Å, and hydrophobic, hydrogen-bond donor, and acceptor properties set to +1.0 [39]. In this framework, five fields were computed: steric, electrostatic, hydrophobic, hydrogen-bond donor, and hydrogen-bond acceptor. Field strength attenuation was modeled using a Gaussian-type distance dependence with an attenuation factor (α) of 0.3.

#### 3.1.3. Model Construction and Internal Validation

The relationship between CoMSIA field descriptors (independent variables) and biological activity (dependent variable) was analyzed using Partial Least Squares (PLS) regression [40]. Model robustness was assessed through cross-validation, applying the leave-one-out (LOO) strategy in conjunction with the SAMPLS algorithm to estimate the cross-validated squared correlation coefficient (q^2^) and to identify the optimal number of components (N). For the non-cross-validated PLS analysis, a column filter value of 2.0 was employed to reduce background noise and improve interpretability.

The internal predictive ability of the model (q^2^) was calculated using Equation (2):(2)$$q^{2}=1-\frac{\sum \left(y_{i}-y_{pred}\right)}{\sum \left(y_{i}-y_{ave}\right)}$$
where *y_i_* is the observed activity of the training set, *y_pred_* is the predicted activity of the training set, and *y_ave_* is the average of the activity of the training set.

#### 3.1.4. External Validation of the CoMSIA Model

External validation was assessed using the predictive squared correlation coefficient (*r*^2^*_pred_*), calculated as:(3)$$r_{pred}^{2}=\frac{SD-PRESS}{SD}$$

SD is defined as the squared deviation of test set activities from the training set mean, and PRESS as the squared difference between observed and predicted test set activities.

Further statistical parameters for external validation included:

Mean absolute error-based coefficient (*r_m_*^2^):(4)$$r_{m}^{2}=r^{2}(1-\left|\sqrt{r^{2}-r_{0}^{2}}\right|)$$

Predictive sum of squared deviations (PRESS):(5)$$PRESS=\;\sum_{i=1}^{n_{EXT}}{\left(y_{i}-{\hat{y}}_{i}\right)}^{2}$$

Sum of squares deviations from training set mean (SSD):(6)$$SSD=\;\sum_{i=1}^{n_{EXT}}{\left(y_{i}-{\bar{y}}_{TRA}\right)}^{2}$$
where *TR* and *EXT* denote the training and external test sets, respectively; *y_i_* are experimental activity values, *ŷ_i_* are predicted values, and *ȳ* is the average of the training set activities. *r*_0_^2^ is the coefficient of determination for regression through the origin.

#### 3.1.5. Two-Dimensional Quantitative Structure-Activity Relationship Model

The 2D-QSAR (Hansch analysis) model was developed using the same compound dataset applied in the 3D-QSAR study, with modifications based on established methodologies [25,41]. Molecular geometries were optimized at the DFT-B3LYP-6-31G(d,p) theory level using Gaussian 09 (Version 7, 1995-09 Gaussian Inc., Wallingford, CT, USA), with the absence of imaginary frequencies confirming true minima [42]. Quantum chemical descriptors—dipole moment (DM), atomic charges (C1–C6, primed analogs, Cα, Cβ, CO), frontier orbital energies (HOMO, LUMO), energy gap (ΔLH), chemical potential (μ), hardness (η), softness (S), and electrophilicity index (ω)—were obtained from the output files and calculated as follows:(7)$$\Delta LH=\left(E_{LUMO}-E_{HOMO}\right)$$(8)$$\mu =\frac{\left(E_{LUMO}+E_{HOMO}\right)}{2}$$(9)$$\eta =\frac{\left(E_{LUMO}-E_{HOMO}\right)}{2}$$(10)$$S=\frac{1}{2\eta }$$(11)$$\omega =\frac{\mu^{2}}{2\eta }$$

Further steric and topological descriptors were calculated using molecular mechanics optimization in ChemDraw version 15.1.0.144 (PerkinElmer Inc., Shelton, CT, USA). These included molecular calculated LogP (CLogP), weight (MW), molecular surface area (MS), the number of rotatable bonds (RT), the number of hydrogen-bond acceptors (HBA) and donors (HBD), molar refractivity (MR), and polar surface area (PS), and molecular volume (MV).

The final 2D-QSAR model was developed via multiple linear regression (MLR), maintaining a 1:5 ratio of descriptors to training set compounds to avoid overfitting. Internal validation of the model was performed using the cross-validated q^2^ metric, as defined in Equation (2).

#### 3.1.6. Evaluation of Physicochemical and Pharmacokinetic Properties

The absorption, distribution, metabolism, and excretion (ADME) properties of the most promising quinoline–chalcone derivatives were evaluated using the SwissADME online platform [43]. The parameters analyzed included the number of atoms with hydrogen-bond acceptor capacity (H-bond acceptors), molar refractivity (MR), octanol/water partition coefficient estimated from solvation free energy in implicit solvents (iLogP), and aqueous solubility predicted directly from molecular structure (ESOL log S), along with the corresponding solubility classification (ESOL class). Additional descriptors comprised predicted gastrointestinal absorption (GI absorption), blood–brain barrier permeability (BBB permeant), and the probability of acting as a P-glycoprotein substrate (P-gp substrate). Furthermore, cytochrome P450 isoenzyme inhibition profiles (CYP2C19, CYP2C9, CYP2D6, and CYP3A4), the number of Lipinski’s rule-of-five violations, and the predicted oral bioavailability according to the Abbott score were also considered.

### 3.2. Synthesis of Compounds

#### 3.2.1. Instruments and Chemicals

Compound structures were characterized through MP, IR, ^1^H- and ^13^C-NMR data. MP was determined with an Electrothermal 9100 SERIES-Digital apparatus (UK). IR spectra were collected on a Thermo Scientific FT-IR (Madison, WI, USA) using KBr disks. ^1^H- and ^13^C-NMR spectra were recorded on a Bruker 600 spectrometer (Fällanden, Switzerland) at 600 and 151 MHz, respectively. The NMR spectra were calibrated using the chemical shift in TMS (δ = 0.0). ^13^C NMR spectra of all compounds were recorded under selective ^19^F decoupling conditions. Chemical shifts (δ) are given in ppm. Elemental analysis (C, H, and N) were performed on an Exeter CE 440 (Coventry, UK) and the results were within ±0.4% of the calculated values. TLC analyses employed DC-Alufolien Kieselgel 60 F254 plates (Merck, St. Louis, MO, USA).

#### 3.2.2. Chemistry

##### Synthesis of 1-(6-chloro-4-(2-fluorophenyl)-2-methylquinolin-3-yl)ethan-1-one (**III**)

A solution of **III-1** (537.8 mg, 2.16 mmol), **III-2** (1091.3 mg, 10.90 mmol), and trifluoroacetic acid (0.15 mL, 0.002 mmol) in ethanol was heated in a 50 mL round-bottom flask. After completion (TLC monitoring), the mixture was diluted with CH_2_Cl_2_ (15 mL), washed with sat. NaHCO_3_ (10 mL) and brine (10 mL), dried (Na_2_SO_4_), and concentrated. The resulting solid was triturated with methanol. Yellow solid, yield 82%, mp: (121–123) °C. IR (cm^−1^): 2963 (C-H), 1706 (C=O), 1491 (C=C_Ar_), 1447 (C=C), 1249 (C-F), 1166 (C-CO), 721 (C-Cl). ^1^H-NMR (CDCl_3_): δ 2.76 (3H, s), 3.39 (3H, s), 7.14–7.16 (1H, m, H-Ar), 7.25 (1H, t, *J* = 7.4 Hz, H-Ar), 7.33 (1H, t, *J* = 7.3 Hz, H-Ar), 7.45 (1H, d, *J* = 2.2 Hz, H-Ar), 7.52–7.54 (1H, m, H-Ar), 7.74 (1H, dd, *J* = 9.1 and 2.0 Hz, H-Ar), 8.04 (1H, d, *J* = 9.0 Hz) ppm. ^13^C-NMR (CDCl_3_): δ 34.4, 40.3, 115.7, 124.2, 124.9, 126.0, 127.9, 129.9, 129.9, 130.2, 130.2, 132.8, 143.8, 147.2, 158.4, 160.0, 162.4, 197.6 ppm.

##### General Procedure for the Synthesis of Quinoline–Chalcones **E001**–**E012**

Compound **III** (313 mg, 1.0 mmol) and aldehyde **IV** (between 127.2 and 271.2 mg, 1.2 mmol) were refluxed in 5 mL of 10% ethanolic NaOH for ~6 h (TLC monitoring, Figure 1). The precipitate was filtered and recrystallized from methanol to afford the final product. Details of the spectroscopic data of compounds **E001**–**E012** are in the Appendix A.

(*E*)-1-(6-chloro-4-(2-fluorophenyl)-2-methylquinolin-3-yl)-3-(3,4-dimetoxyphenyl)prop-2-en-1-one (**E001**).

Yellow solid, yield 64%, mp (168–170) °C. IR (cm^−1^): υ 3017, 2954, 2815, 1605, 1496, 1117, 1057, 728, 678. ^1^H-NMR (DMSO-d_6_): δ 2.69 (3H, s, CH_3_-Ar), 3.88 (3H, s, OCH_3_), 3.90 (3H, s, OCH_3_), 6.91 (1H, d, *J* = 15.9 Hz, CH=CH), 7.03 (1H, s, H-Ar), 7.09 (2H, s, H-Ar), 7.16 (1H, t, *J* = 7.1 Hz, H-Ar), 7.23 (1H, t, *J* = 7.1 Hz, H-Ar), 7.35 (1H, d, *J* = 16.1 Hz, CH=CH), 7.37 (1H, s, H-Ar), 7.50 (1H, t, *J* = 7.2 Hz, H-Ar), 7.63 (1H, s, H-Ar), 7.85 (1H, d, *J* = 7.2 Hz, H-Ar), 8.15 (1H, d, *J* = 7.8 Hz, H-Ar) ppm. ^13^C-NMR (DMSO-d_6_): δ 23.5, 55.9, 56.0, 111.2, 112.3, 115.8, 121.9, 122.4, 123.6, 123.9, 124.4, 125.3, 127.0, 127.5, 128.9, 129.5, 130.7, 131.2, 136.9, 143.6, 148.7, 150.0, 150.5, 157.7, 158.8, 159.4, 190.8 ppm.

(*E*)-3-(benzo[*d*][1,3]dioxol-5-yl)1-(6-chloro-4-(2-fluorophenyl)-2-methylquinolin-3-yl)prop-2-en-1-one (**E002**)

Beige solid, yield 79%, mp (180–181) °C. IR (cm^−1^): υ 3049, 2890, 1589, 1505, 1483, 1369, 1245, 1037, 827, 776, 700. ^1^H-NMR (DMSO-d_6_): δ 2.72 (3H, s, CH_3_-Ar), 6.01 (2H, s, OCH_2_O), 6.50 (1H, d, *J* = 18 Hz, CH=CH), 6.82 (1H, dd, *J* = 8.2, 1.8 Hz, H-Ar), 6.89 (1H, s, H-Ar), 7.09 (1H, d, *J* = 18 Hz, CH=CH), 7.11 (1H, d, *J* = 8.2 Hz, H-Ar), 7.23–7.25 (1 H, m, H-Ar), 7.28 (1H, s, H-Ar), 7.43–7.47 (2H, m, H-Ar), 7.70 (1H, d, *J* = 1.8 Hz, H-Ar), 8.09 (1H, d, *J* = 1.8 Hz, H-Ar). ^13^C-NMR (DMSO-d_6_): δ 22.6, 101.2, 107.1, 108.7, 115.4, 115.6, 123.5, 124.3, 124.3, 124.8, 125.4, 125.7, 125.7, 126.2, 126.3, 126.4, 127.0, 127.1, 128.6, 129.4, 129.7, 130.8, 132.1, 132.2, 136.8, 136.9, 144.2, 148.1, 148.4, 148.4, 157.7, 160.6, 162.5, 190.6.

(*E*)-1-(6-chloro-4-(2-fluorophenyl)-2-methylquinolin-3-yl)-3-(3,4-difluorophenyl)prop-2-en-1-one (**E003**)

White solid, yield 73%, mp (188–189) °C. IR (cm^−1^): υ 2998, 2816, 1693, 1458, 1326, 1215, 930, 728, 542. ^1^H-NMR (DMSO-d_6_): δ 2.70 (3H, s, CH_3_-Ar), 6.90 (1H, d, *J* = 15.9 Hz, CH=CH), 7.22–7.23 (2H, m, H-Ar), 7.26–7.29 (3H, m, H-Ar), 7.34–7.35 (1H, m, H-Ar), 7.39 (1H, d, *J* = 16.1 Hz, CH=CH), 7.51–7.53 (1H, m, H-Ar), 7.64 (1H, s, H-Ar), 7.85 (1H, d, *J* = 9.0 Hz, H-Ar), 8.14 (1H, d, *J* = 6.6 Hz, H-Ar) ppm. ^13^C-NMR (DMSO-d_6_): δ 23.6, 115.93, 117.4, 122.1, 123.6, 123.9, 124.4, 125.3, 127.5, 127.8, 129.5, 130.2, 130.7, 131.2, 131.5, 136.9, 144.1, 148.7, 150.3, 152.1, 153.9, 157.7, 158.8, 159.4, 191.4 ppm. Anal Calculated for C_25_H_15_ClF_3_NO: C 68.58, H 3.45, Cl 8.10, F 13.02, N 3.20, O 3.65. Found: C 68.85, H 3.76, Cl 7.91, F 12.90, N 3.48, O 3.29.

(*E*)-1-(6-chloro-4-(2-fluorophenyl)-2-methylquinolin-3-yl)-3-(3,4,5-trimetoxyphenyl)prop-2-en-1-one (**E004**)

White solid, yield 79%, mp (149–151) °C. IR (cm^−1^): υ 3019, 2983, 1636, 1469, 1128, 729, 636. ^1^H NMR (DMSO-d_6_): δ 2.72 (3H, s, CH_3_-Ar), 3.84 (3H, s, OCH_3_), 3.90 (6H, s, OCH_3_), 6.80 (1H, d, *J* = 15.8 Hz, CH=CH), 6.82 (2H, s, H-Ar), 7.07–7.10 (1H, m, H-Ar), 7.14–7.17 (1H, m, H-Ar), 7.29 (1H, d, *J* = 15.6 Hz, CH=CH), 7.34–7.36 (1H, m, H-Ar), 7.47 (1H, t, *J* = 7.1 Hz, H-Ar), 7.63 (1H, s, H-Ar), 7.86 (1H, d, *J* = 6.3 Hz, H-Ar), 8.15 (1H, d, *J* = 7.8 Hz) ppm. ^13^C NMR (DMSO-d_6_): δ 23.1, 55.8, 56.2, 60.7, 105.5, 115.2, 115.9, 120.63123.6, 123.9, 124.4, 125.4, 127.5, 128.5, 129.2, 129.5, 129.8, 130.5, 130.7, 136.8, 140.3, 142.3, 152.4, 153.4, 157.6, 158.3, 159.0, 191.0 ppm.

(*E*)-1-(6-chloro-4-(2-fluorophenyl)-2-methylquinolin-3-yl)-3-(4-(trifluoromethyl)phenyl)prop-2-en-1-one (**E005**)

Yellow solid, yield 52%, mp (197–199) °C. IR (cm^−1^): υ 2995, 2852, 1691, 1643, 1425, 1357, 1237, 699, 635. ^1^H-NMR (DMSO-d_6_): δ 2.71 (3H, s, CH_3_-Ar), 6.99 (1H, d, *J* = 16.1 Hz, CH=CH), 7.23–7.26 (1H, m, H-Ar), 7.27–7.30 (1H, m, H-Ar), 7.33–7.37 (1H, m, H-Ar), 7.39 (1H, d, *J* = 16.0 Hz, CH=CH), 7.46 (1H, d, *J* = 6.1 Hz, H-Ar), 7.51–7.53 (1H, m, H-Ar), 7.64 (1H, s, H-Ar), 7.73 (1H, d, *J* = 5.8 Hz, H-Ar), 7.85 (1H, d, *J* = 5.9 Hz, H-Ar), 8.14 (1H, d, *J* = 6.1 Hz, H-Ar) ppm. ^13^C-NMR (DMSO-d_6_): δ 21.7, 115.4, 115.6, 123.0, 124.3, 124.3, 125.2, 125.4, 125.7, 125.7, 125.9, 125.9, 125.9, 126.0, 126.2, 126.3, 127.0, 127.1, 128.6, 129.1, 129.2, 129.2, 129.4, 130.4, 130.6, 130.8, 132.2, 132.2, 135.2, 136.8, 144.3, 148.4, 157.7, 160.6, 162.5, 190.7 ppm. Anal Calculated for C_26_H_16_ClF_4_NO: C 66.46, H 3.43, Cl 7.54, F 16.17, N 2.98, O 3.41. Found: C 66.29, H 3.76, Cl 7.38, F 15.88, N 2.64, O 3.70.

(*E*)-1-(6-chloro-4-(2-fluorophenyl)-2-methylquinolin-3-yl)-3-(3,4-dichlorophenyl)prop-2-en-1-one (**E006**)

White solid, yield 55%, mp (181–183) °C. IR (cm^−1^): υ 3049, 2890, 1589, 1505, 1483, 1369, 1245, 1037, 827, 776, 700. ^1^H-NMR (DMSO-d_6_): δ 2.70 (3H, s, CH_3_-Ar), 7.37–7.43 (3H, m, H-Ar), 7.51 (2H, d, *J* = 7.1 Hz, H-Ar), 7.52 (1H, d, *J* = 16.2 Hz, CH=CH), 7.64 (1H, d, *J* = 16.1Hz, CH=CH), 7.66 (1H, s, H-Ar), 7.71–7.72 (1H, m, H-Ar), 7.82 (1H, dd, *J* = 9.3 Hz, 2.3 Hz, H-Ar), 8.13 (1H, d, *J* = 9.7 Hz, H-Ar), 8.24 (1H, d, *J* = 6.1 Hz) ppm. ^13^C-NMR (DMSO-d_6_): δ 23.6, 115.4, 115.6, 123.5, 123.6, 124.3, 124.3, 125.4, 125.7, 124.7, 126.2, 127.0, 127.1, 128.6, 128.6, 129.4, 130.6, 130.8, 131.0, 132.2, 132.2, 133.9, 134.4, 135.4, 136.8, 136.9, 142.9, 148.4, 157.7, 160.6, 162.5, 190.6 ppm. Anal Calculated for C_25_H_15_Cl_3_FNO: C 63.79, H 3.21, Cl 22.59, F 4.04, N 2.98, O 3.40. Found: C 63.91, H 3.47, Cl 22.35, F 4.16, N 3.29, O 3.59.

(*E*)-1-(6-chloro-4-(2-fluorophenyl)-2-methylquinolin-3-yl)-3-(4-metoxyphenyl)prop-2-en-1-one (**E007**)

Yellow solid, yield 70%, mp (187–188) °C. IR (cm^−1^): υ 3067, 2945, 1663, 1611, 1354, 1285, 732, 699. ^1^H-NMR (DMSO-d_6_): δ 2.73 (3H, s, CH_3_-Ar), 3.84 (3H, s, OCH_3_), 6.55 (1H, d, *J* = 18.0 Hz, CH=CH), 6.88 (2H, d, *J* = 6.0 Hz, H-Ar), 7.09 (1H, d, *J* = 6.0 Hz, H-Ar), 7.15 (1H, d, *J* = 18.0 Hz, CH=CH), 7.28–7.29 (1H, m, H-Ar), 7.30 (1H, s, H-Ar), 7.34 (2H, d, *J* = 6.0 Hz, H-Ar), 7.41–7.44 (2H, m, H-Ar), 7.72 (1H, dd, *J* = 1.2, 6.0 Hz, H-Ar), 8.10 (1H, d, *J* = 6.0 Hz, H-Ar). ^13^C-NMR (DMSO-d_6_): δ 21.7, 55.4, 114.5, 115.4, 115.6, 123.5, 123.6, 124.3, 124.3, 125.4, 125.7, 125.7, 126.3, 126.4, 126.4, 127.0, 127.1, 128.6, 128.6, 129.4, 130.3, 130.8, 132.2, 132.2, 136.8, 136.9, 144.4, 148.4, 157.7, 160.6, 161.1, 162.5, 190.7 ppm.

(*E*)-1-(6-chloro-4-(2-fluorophenyl)-2-methylquinolin-3-yl)-3-(6-metoxynapthalen-2-yl)prop-2-en-1-one (**E008**)

Yellow solid, yield 80%, mp (199–201) °C. IR (cm^−1^): υ 2996, 2816, 1641, 1529, 1415, 1058, 710, 679. ^1^H-NMR (DMSO-d_6_): δ 2.73 (3H, s, CH_3_-Ar), 3.87 (3H, s, OCH_3_), 6.94 (1H, s, H-Ar), 6.97 (1H, d, *J* = 15.6 Hz, CH=CH), 7.23 (2H, d, *J* = 9.0 Hz, H-Ar), 7.31–7.36 (2H, m, H-Ar), 7.40 (1H, d, *J* = 9.0 Hz, H-Ar), 7.42 (1H, d, *J* = 15.9 Hz, CH=CH), 7.53 (1H, t, *J* = 7.1 Hz) H-Ar), 7.65 (1H, s, H-Ar), 7.69 (2H, d, *J* = 7.2 Hz, H-Ar), 7.76 (1H, d, *J* = 7.8 Hz, H-Ar), 7.86 (1H, d, *J* = 8.0 Hz, H-Ar), 8.13 (1H, d, *J* = 7.2 Hz, H-Ar) ppm. ^13^C-NMR (DMSO-d_6_): δ 23.6, 55.6, 106.0, 115.8, 115.9, 119.8, 121.9, 122.1, 123.6, 124.4, 126.0, 127.5, 127.8, 128.0, 128.9, 129.6, 130.2, 130.7, 131.1, 131.2, 136.9, 137.8, 144.8, 148.6, 157.7, 158.8, 159.4, 191.4 ppm.

(*E*)-1-(6-chloro-4-(2-fluorophenyl)-2-methylquinolin-3-yl)-3-oxoprop-1en-1-yl)phenyl benzoate (**E009**)

Pink solid, yield 66%, mp (179–181) °C. IR (cm^−1^): υ 3000, 2900, 1721, 1683, 1566, 1468, 1331, 1259, 1037, 833, 708. ^1^H-NMR (DMSO-d_6_): δ 2.69 (3H, s, CH_3_-Ar), 6.95 (1H, d, *J* = 15.0 Hz, CH=CH), 7.19 (2H, d, *J* = 7.8 Hz, H-Ar), 7.22–7.23 (1H, m, H-Ar), 7.27–7.30 (1H, m, H-Ar), 7.35–7.38 (1H, m, H-Ar), 7.38 (1H, d, *J* = 15.0 Hz, CH=CH), 7.51–7.53 (1H, m, H-Ar), 7.59–7.60 (3H, m, H-Ar), 7.65 (1H, s, H-Ar), 7.66 (2H, d, *J* = 9.6 Hz, H-Ar), 7.86 (1H, d, *J* = 7.2 Hz, H-Ar), 8.12–8.16 (2H, m, H-Ar), 8.16 (1H, d, *J* = 7.2 Hz, H-Ar) ppm. ^13^C-NMR (DMSO-d_6_): δ 23.4, 115.8, 121.6, 121.9, 123.6, 123.9, 124.4, 125.2, 127.5, 127.8, 129.0, 129.4, 129.5, 129.6, 130.0, 130.7, 131.2, 131.2, 133.4, 136.8, 137.1, 143.3, 148.7, 151.9, 157.7, 158.8, 158.8, 159.4, 164.9, 191.4 ppm.

(*E*)-1-(6-chloro-4-(2-fluorophenyl)-2-methylquinolin-3-yl)-3-(4-cholrophenyl)prop-2-en-1-one (**E010**)

White solid, yield 63%, (278–279) °C. IR (cm^−1^): υ 3018, 2949, 2856, 1680, 1627, 1365, 968, 729, 642, 593. ^1^H-NMR (DMSO-d_6_): δ 2.72 (3H, s, CH_3_-Ar), 6.81 (1H, d, *J* = 16.1 Hz, CH=CH), 7.21–7.28 (2H, m, H-Ar), 7.33–7.36 (1H, m, H-Ar), 7.39 (1H, d, *J* = 16.1 Hz, CH=CH), 7.39 (2H, d, *J* = 6.2 Hz, H-Ar), 7.57 (2H, d, *J* = 6.0 Hz, H-Ar), 7.67–7.69 (2H, m, H-Ar), 7.85 (1H, d, *J* = 5.9 Hz, H-Ar), 8.14 (1H, d, *J* = 6.0 Hz) ppm. ^13^ C-NMR (DMSO-d_6_): δ 22.2, 115.6, 115.7, 124.2, 124.2, 124.9, 126.0, 127.9, 128.3, 129.9, 130.2, 130.3, 130.4, 132.8, 147.3, 158.4, 160.1, 162.4, 197.6 ppm.

(*E*)-1-(6-chloro-4-(2-fluorophenyl)-2-methylquinolin-3-yl)-3-(3,5-difluorophenyl)prop-2-en-1-one (**E011**)

White solid, yield 87%, mp (176–178) °C. IR (cm^−1^): υ 2947, 2839, 1625, 1367, 1350, 731, 676. ^1^H-NMR (DMSO-d_6_): δ 2.71 (3H, s, CH_3_-Ar), 6.98 (1H, d, *J* = 15.0 Hz, CH=CH), 6.99 (1H, s, H-Ar), 7.06 (1H, s, H-Ar), 7.07 (1H, s, H-Ar), 7.21–7.23 (1H, m, H-Ar), 7.29–7.30 (1H, m, H-Ar), 7.34–7.36 (1H, m, H-Ar), 7.42 (1H, d, *J* = 15.0 Hz, CH=CH), 7.51–7.53 (1H, m, H-Ar), 7.65 (1H, s, H-Ar), 7.86 (1H, d, *J* = 8.4 Hz, H-Ar), 8.15 (1H, d, *J* = 7.8 Hz, H-Ar) ppm. ^13^C-NMR (DMSO-d_6_): δ 24.9, 102.6, 112.9, 115.9, 121.9, 123.6, 123.9, 124.4, 125.4, 127.5, 127.5, 129.4, 130.7, 131.2, 136.3, 136.8, 143.8, 148.7, 157.7, 158.8, 159.4, 162.8, 164.5, 190.8 ppm. Anal Calculated for C_25_H_15_ClF_3_NO: C 68.58, H 3.45, Cl 8.10, F 13.02, N 3.20, O 3.65. Found: C 68.24, H 3.73, Cl 7.81, F 13.24, N 3.38, O 3.88.

(*E*)-1-(6-chloro-4-(2-fluorophenyl)-2-methylquinolin-3-yl)-3-phenylprop-2-en-1-one (**E012**)

Beige solid, yield 83%, mp (154–156) °C. IR (cm^−1^): υ 2935, 2874, 1711, 1643, 1427, 973, 733, 696. ^1^H-NMR (DMSO-d_6_): δ 2.69 (3H, s, CH_3_-Ar), 6.54 (2H, d, *J* = 18.0 Hz, CH=CH), 6.85 (2H, d, *J* = 7.2 Hz, H-Ar), 7.11 (1H, d, *J* = 7.2 Hz, H-Ar), 7.16 (1H, d, *J* = 18.0 Hz, CH=CH), 7.24–7.27 (3H, m, H-Ar), 7.36 (2H, d, *J* = 7.1 Hz, H-Ar), 7.43–7.47 (2H, m, H–Ar), 7.73 (1H, d, *J* = 7.2 Hz, H-Ar), 8.10 (1H, d, *J* = 7.0 Hz, H-Ar) ppm. ^13^C-NMR (DMSO-d_6_): δ 21.6, 115.4, 115.6, 123.5, 123.6, 124.3, 124.3, 125.4, 125.7, 126.3, 126.4, 126.8, 127.0, 127.1, 128.6, 128.9, 129.4, 130.0, 130.8, 132.2, 132.2, 135.6, 136.8, 138.9, 144.5, 148.4, 157.7, 160.6, 162.5, 190.7 ppm.

#### 3.2.3. Biology

##### Parasites

Promastigotes of *Leishmania amazonensis* (IFLA/BR/67/PH8) were maintained in M199 medium enriched with 20% FBS, 0.1% hemin (25 mg/mL prepared in 0.1 N NaOH), 10 mM adenine, and 5% penicillin/streptomycin, at pH 7.4 and 26 °C. For MTT assays, stationary-phase cultures obtained after 7 days of growth were used. Before experimentation, the parasites were washed three times with 0.01 M PBS, and 3 × 10^6^ promastigotes were seeded per well for incubation.

##### Viability Assay

The effect of the compounds was assessed by colorimetric viability assay, with 3-[4,5-dimethyl-2-thiazolyl]-2,5-diphenyl-2H-tetrazolium bromide, MTT. The promastigotes were incubated at 3 × 10^6^ parasites per well in the presence of different compounds at different concentrations (3.66, 7.33, 14.65, and 29.30 µM) for 24 h after 5 mg/mL of MTT (Sigma-Aldrich, St. Louis, MO, USA) were added for 2 h, and then the reaction was stopped with 30% of SDS (sodium dodecyl sulfate). Positive control of treatment was performed with 2 μM of amphotericin B (Sigma-Aldrich). The optical density was determined in a plate reader (Multiskan EX—Labsystem, Thermo Fisher Scientific, Waltham, MA, USA) of 595 nm. Results were expressed as the mean percentage viability of treated parasites compared with untreated parasites.

##### Statistical Analyses

All the experiments were performed in three independent replicates, and each was carried out in technical triplicate. The statistical significance was assessed using GraphPad Prism 6.0.1 (Dotmatics, Boston, MA, USA) by ANOVA followed Tukey’s Multiple Comparison Test.

## 4. Conclusions

In the current study, quantitative structure–activity relationship (QSAR) models were developed using 2D-QSAR (focused on physicochemical properties) and 3D-QSAR (focused on conformational properties) analyses, based on the chalcone pharmacophore. From a statistical perspective, the resulting models are robust, meeting the criteria for both internal and external validation. The 2D-QSAR model revealed that the properties influencing the inhibitory activity against *L. amazonensis* include the Lowest Unoccupied Molecular Orbital (LUMO) and the global electrophilicity index (ω). In the case of the 3D-QSAR model, the CoMSIA steric and hydrogen-bond acceptor (CoMSIA-SA) analysis indicated a contribution of 77.5% from the steric field and 22.5% from the hydrogen-bond acceptor field.

To support the validity of the developed models, twelve quinoline–chalcone hybrid compounds (**E001**–**E012**) were synthesized in two steps, with overall yields ranging from 43% to 71%. These compounds were evaluated for their inhibitory activity against *L. amazonensis*, with IC_50_ values ranging from 2.23 to 13.27 μM. Four derivatives displayed activities comparable to the reference drug, amphotericin B. A comparison between experimental and predicted results from the 2D- and 3D-QSAR models showed good predictability (ΔpIC_50_ < 1.0).

Additionally, the physicochemical and pharmacokinetic properties of the four most active compounds (**E003**, **E005**, **E006**, and **E011**) revealed low water solubility, low intestinal absorption, low probability of crossing the blood–brain barrier, and minimal potential for inhibition of major cytochrome enzymes. These characteristics suggest high biocompatibility.

Overall, these findings demonstrate that the developed 2D- and 3D-QSAR models possess strong predictive power and could serve as a rational basis for the design of a new generation of *L. amazonensis* growth inhibitors based on the chalcone pharmacophore.

## Data Availability

The original contributions presented in this study are included in the article/Appendix A. Further inquiries can be directed to the corresponding authors.

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
