# Peer review of "Design of Hybrid Quinoline–Chalcone Compounds Against Leishmania amazonensis Based on Computational Techniques: 2D- and 3D-QSAR with Experimental Validation"

_pharmaceuticals, 2025, doi:10.3390/ph18101567_

Round 1
Reviewer 1 Report
Comments and Suggestions for Authors
1. If the compound is assigned a code in the text, it is no longer necessary to refer to the compound by its full name; use the code assigned.
2. In the chemical synthesis results, briefly describe the NMR results. I don't think this is necessary since the results for each compound are presented in the methodology, and if you are going to describe the resonance, you should also describe the IR.
3. In the ADME analysis, add the consultation date and citations.
4. Could the compounds present hepatotoxicity? Predictive analysis could complement these results.
5. Table 3 does not present the meanings of the abbreviations in the table footnote.
6. In the ADME analysis, the methodology does not describe the abbreviations.
7. In the methodology where the IR spectra are described, the corresponding signal is not indicated.
8. The IR spectra were not included in the supplementary material.
Author Response
Thank you very much for the comments regarding our manuscript entitled “Design of hybrid quinoline-chalcone compounds against Leishmania amazonensis based on computational techniques: 2D- and 3D-QSAR with experimental validation”. We have carefully addressed all the points raised and have incorporated the suggested revisions into the manuscript and supplementary material. All modifications are highlighted in yellow for ease of identification.
We sincerely thank the reviewers for their valuable feedback, which has significantly improved our manuscript. We have carefully addressed all comments and hope the revised version meets their expectations.
Reviewer 1 comments:
- If the compound is assigned a code in the text, it is no longer necessary to refer to the compound by its full name; use the code assigned.
Response: We appreciate your suggestion for improving the readability of our research. In the results and discussion section, we have removed the full IUPAC names of the compounds, using only the compound numbers. However, we have retained the IUPAC names in the materials and methods section.
- In the chemical synthesis results, briefly describe the NMR results. I don't think this is necessary since the results for each compound are presented in the methodology, and if you are going to describe the resonance, you should also describe the IR.
Response: We appreciate your suggestion. The indicated section regarding the results of the 1H-NMR for the quinoline-chalcone hybrid compounds identification was deleted to improve the read flow.
- In the ADME analysis, add the consultation date and citations.
Response: We thank you for your recommendation. We added the requested information in the references section.
- Could the compounds present hepatotoxicity? Predictive analysis could complement these results.
Response: We sincerely thank the reviewer for highlighting the important issue of potential hepatotoxicity associated with leishmanicidal compounds. We fully agree that assessing liver toxicity is a critical step prior to any preclinical in vivo evaluation, particularly since our current findings are based exclusively on in vitro assays using promastigotes.
Although our results demonstrate significant in vitro activity, we acknowledge that cytotoxicity in hepatocytes and potential liver injury cannot be reliably inferred from protozoan data. Drug-induced liver injury (DILI) is a multifactorial process influenced by various mechanisms, including metabolic activation, reactive metabolite formation, transporter interactions, mitochondrial impairment, and inflammatory responses. Therefore, dedicated experimental models are required to accurately predict hepatotoxic potential.
We strongly believe that complementary studies will be indispensable before proceeding to murine trials, and we are committed to addressing this crucial aspect in our future research. Nevertheless, a preliminary approximation of possible liver toxicity can be inferred from the ADME-related parameters summarized in Table 3. Based on the predicted interactions of the designed compounds with CYP450 isoenzymes, most of them are not classified as CYP inhibitors. This suggests that these compounds are unlikely to interfere with or inhibit cytochrome P450 enzymatic activity, which is predominantly expressed in the liver and plays a major role in xenobiotic metabolism.
- Table 3 does not present the meanings of the abbreviations in the table footnote.
Response: We appreciate your observation. The meaning of each ADME property was added as a footnote in Table 3.
- In the ADME analysis, the methodology does not describe the abbreviations
Response: We thank you for your suggestion. The prediction of ADME properties was added to the methodology, describing all abbreviations used in Table 3 in the Materials and Methods section (lines 463-474).
- In the methodology where the IR spectra are described, the corresponding signal is not indicated.
Response: We appreciate your keen observation. Indeed, we forgot to add the information in the first submission. The information has been added to the materials and methods section (lines 479-480).
- The IR spectra were not included in the supplementary material.
Response: We appreciate your keen observation. Indeed, we forgot to add the information in the first submission. The information has been added to the supplementary material section.
Reviewer 2 Report
Comments and Suggestions for Authors
The manuscript describes computations–driven design of compounds, active against Leishmania amazonensis. It reports construction of two predictive QSAR models and design of quinoline-chalcone hybrid molecules basing on obtained structural information, followed by synthesis and bioevaluation of the compounds with predicted activity. As a result, several compounds with desired activity were found and quality of the elaborated predictive models was estimated as acceptable.
The manuscript can be accepted for publication after the following remarks be addressed.
1) What are the values of predicted activity for non-active compounds E001,E008,E010? Do they significantly differ from the values for active ones?
2) Errors for IC50 values (13.27±...) and IC50 for control compound should be given in Table 2.
3) Viability Assay (section 3.2.3.2) is described for only one concentration.
4) For all novel compounds HRMS or elemental analysis data should be given.
5) In 1H NMR spectra registered in DMSO-d6 no solvent residual signals are observed at 2.50 ppm. Either it is mistakenly reported that 1H NMR spectra (except for compound III) were registered in DMSO-d6, or chemical shifts are given with a wrong reference.
6) In 13C NMR spectra of compounds containing fluorine atoms there are observed spin–spin coupling constants C-F (about 250 Hz for 1JCF or less for more distant nuclei). Either it should be reflected in NMR spectra description or special remark should be done if 13C NMR spectra were registered in fluorine-decoupled mode.
7) Description of 1H NMR spectra is not everywhere correct. For example, in 1H NMR spectrum of compound III signal at 7.15 ppm is not a triplet, as six lines are clearly observed. In this case and all similar cases the signal should be described as multiplet and characterized (as for all multiplets should be done) with a diapason of chemical shifts, not the centre.
8) In description of 13C NMR spectra chemical shifts should be given to one decimal place (23.5 not 23.54 ppm).
9) page 3, last paragraph – “hybridization of two or more molecular fragments”. It would be better to rephrase as hybridization is associated more with orbitals than with hybrid molecules.
10) page 1, line 2 – it would be better to mention here, where equation 4 is.
11) In the title of the article “of” after "Design..." seems omitted.
12) What do references cited in Table S1 correspond to?
Author Response
Thank you very much for the comments regarding our manuscript entitled “Design of hybrid quinoline-chalcone compounds against Leishmania amazonensis based on computational techniques: 2D- and 3D-QSAR with experimental validation”. We have carefully addressed all the points raised and have incorporated the suggested revisions into the manuscript and supplementary material. All modifications are highlighted in yellow for ease of identification.
We sincerely thank the reviewers for their valuable feedback, which has significantly improved our manuscript. We have carefully addressed all comments and hope the revised version meets their expectations.
Reviewer 2 comments:
The manuscript describes computations–driven design of compounds, active against Leishmania amazonensis. It reports construction of two predictive QSAR models and design of quinoline-chalcone hybrid molecules basing on obtained structural information, followed by synthesis and bioevaluation of the compounds with predicted activity. As a result, several compounds with desired activity were found and quality of the elaborated predictive models was estimated as acceptable.
The manuscript can be accepted for publication after the following remarks be addressed.
1) What are the values of predicted activity for non-active compounds E001, E008, E010? Do they significantly differ from the values for active ones?
Response: Thank you for your question. Compounds E001, E008, and E010 show the activities listed in Table R1.
Table R1: Inhibition activity on L. amazonensis of compounds E001, E008, and E010.
|
Compound |
pIC50 |
|
|
CoMSIA-SA |
2D-QSAR |
|
|
E001 |
5.975 |
7.236 |
|
E008 |
5.890 |
7.040 |
|
E010 |
5.690 |
6.579 |
Although activity predictions for synthesized compounds on L. amazonensis are higher than in Table S2, we could not experimentally measure inhibitory activity due to their insolubility.
To show compound insolubility, we added the category “In: insoluble” at the bottom of Table 2, as in the table body.
2) Errors for IC50 values (13.27±...) and IC50 for control compound should be given in Table 2.
Response: We thank you for your keen observation. The IC50 values were updated, including the ± SD.
3) Viability Assay (section 3.2.3.2) is described for only one concentration.
Response: We appreciate your suggestion. All concentrations tested were added in line 644, corresponding to the section of the viability assay.
4) For all novel compounds HRMS or elemental analysis data should be given.
Response: We thank you for your recommendation. The experiments on HRMS were not possible; however, to ensure the percentage of each atom in the molecular structure, an elemental analysis was performed. Elemental determinations were carried out for the active compounds (E003, E005, E006, E011). Added to the manuscript in lines 483 and 485: Elemental analysis (C, H, and N) were performed on an Exeter CE 440 (Conventry, UK), and the results were within ±0.4% of the calculated values
5) In 1H NMR spectra registered in DMSO-d6 no solvent residual signals are observed at 2.50 ppm. Either it is mistakenly reported that 1H NMR spectra (except for compound III) were registered in DMSO-d6, or chemical shifts are given with a wrong reference.
Response: We thank you for your keen observation. The 1H-NMR and 13C-NMR spectra were calibrated using the chemical shift of TMS (δ= 0.0). This information was added in the material and method section (lines 481-482).
6) In 13C NMR spectra of compounds containing fluorine atoms there are observed spin–spin coupling constants C-F (about 250 Hz for 1JCF or less for more distant nuclei). Either it should be reflected in NMR spectra description or special remark should be done if 13C NMR spectra were registered in fluorine-decoupled mode.
Response: We appreciate your suggestion to improve our manuscript. We added to the manuscript in lines 482 and 483: ¹³C NMR spectra of all compounds were recorded under selective ¹⁹F decoupling conditions.
7) Description of 1H NMR spectra is not everywhere correct. For example, in 1H NMR spectrum of compound III signal at 7.15 ppm is not a triplet, as six lines are clearly observed. In this case and all similar cases the signal should be described as multiplet and characterized (as for all multiplets should be done) with a diapason of chemical shifts, not the centre.
Response: We appreciate your observation to improve the spectroscopic information of the synthetic compounds. The information was corrected in the main manuscript between lines 490-631
8) In description of 13C NMR spectra chemical shifts should be given to one decimal place (23.5 not 23.54 ppm).
Response: We appreciate your keen observation. The requested change was made.
9) page 3, last paragraph – “hybridization of two or more molecular fragments”. It would be better to rephrase as hybridization is associated more with orbitals than with hybrid molecules.
Response: We thank you for your recommendation. The sentence was changed to: “…the development of highly active compounds is the combination of two or more molecular fragments in a unique structure, obtaining a compound known as a hybrid compound” in lines 99 - 101.
10) page 1, line 2 – it would be better to mention here, where equation 4 is.
Response: We thank you for your suggestion. For a better understanding of the results, we have moved the table containing the validation equation results to the supplementary material (Table S5) and have cited it appropriately in the main manuscript text.
11) In the title of the article “of” after "Design..." seems omitted.
Response: We appreciate your keen observation. The article “of” was added in the title.
12) What do references cited in Table S1 correspond to?
Response: We thank you for your question. The references cited in Table S1 correspond to the articles consulted to obtain the structure and biological activity of L. amazonensis in order to construct the dataset used to develop the 3D- and 2D-QSAR models.
Round 2
Reviewer 2 Report
Comments and Suggestions for Authors
I appreciate the athours' attention to the comments and suggestions.
As all the necessary revisions have been done, including providing elemental analysis data for key compounds, I recommend to accept the manuscript for publication.